# Exploiting nonaqueous self-stratified electrolyte systems toward large-scale energy storage

Zhenkang Wang[1,4], Haoqing Ji ®[1,4], Jinqiu Zhou ®[2], Yiwei Zheng[1], Jie Liu ®[2], Tao Qian[2,3] & Chenglin Yan ®[1,3] ✉

Biphasic self-stratified batteries (BSBs) provide a new direction in battery philosophy for large-scale energy storage, which successfully reduces the cost and simplifies the architecture of redox flow batteries. However, current aqueous BSBs have intrinsic limits on the selection range of electrode materials and energy density due to the narrow electrochemical window of water. Thus, herein, we develop nonaqueous BSBs based on Li-S chemistry, which deliver an almost quadruple increase in energy density of 88.5 Wh L$^{-1}$ as compared with the existing aqueous BSBs systems. In situ spectral characterization and molecular dynamics simulations jointly elucidate that while ensuring the mass transfer of Li$^+$, the positive redox species are strictly confined to the bottom-phase electrolyte. This proof-of-concept of Li-S BSBs pushes the energy densities of BSBs and provides an idea to realize massive-scale energy storage with large capacitance.

Currently, the energy crisis caused by the extensive depletion of fossil fuels calls for the development of renewable energy[1]. However, most green energies, such as photovoltaic and hydroelectric power, do not deliver on demand but rather intermittently and on availability, which cannot easily be directly plugged into the electrical grid[2]. Improved technologies to store electrical energy from intermittent renewable sources at random times and better balance the electricity supply and demand have become increasingly urgent. In contrast to conventional energy storage devices built from single batteries, redox flow batteries (RFBs) can decouple the energy and power ratings by storing active species in individual tanks and pumping isolated electrolytes into reaction devices[3,4]. Due to this unique characteristic, RFBs have drawn substantial attention in recent decades. Within RFBs, at least one active species is dissolved in the electrolyte to enable the flow. For example, conventional all-vanadium RFBs rely on dissolved vanadium ions with different valences in catholytes and anolytes[5]. Given the nature of anode and cathode species capable of spontaneous reactions, an ion-exchange membrane is indispensable to prevent active compound

migration and crossover while inhibiting the consumption of charge carriers. However, the costs of the current ion-exchange membranes are too high. They cannot isolate the catholyte and anolyte for a long time, which sets an obstacle for massive commercialization of RFBs[6,7].

Biphasic self-stratified batteries (BSBs) provide a new battery philosophy for RFBs. In 2017, Marcilla et al. utilized the difference in the extraction partition coefficient between water and an ionic liquid, first realizing membrane-free aqueous BSBs. The active species were constrained in two different immiscible electrolyte phases due to the solubility difference[8]. Shortly after, other self-stratified aqueous electrolyte systems were developed based on this philosophy, and various active materials were developed to couple with this new architecture[9–11]. Recently, Shen et al. improved this system by using a stirred electrode to reduce polarization and further improve the overall energy density, which demonstrates the potential of BSBs for large-scale applications[12]. These BSBs based on biphasic systems successfully eliminate the dependence on ion-exchange membranes for RFBs. However, water is necessary to implement the biphasic

[1]Key Laboratory of Core Technology of High Specific Energy Battery and Key Materials for Petroleum and Chemical Industry, College of Energy, Soochow University, Suzhou 215006, China. [2]College of Chemistry and Chemical Engineering, Nantong University, Seyuan 9, Nantong 226000, China. [3]Light Industry Institute of Electrochemical Power Sources, Suzhou 215600, China. [4]These authors contributed equally: Zhenkang Wang, Haoqing Ji. ✉e-mail: c.yan@suda.edu.cn

electrolyte systems applied in existing BSBs. The narrow electrochemical window of water prevents aqueous BSBs from working at high voltages[13]. On the other hand, most electrode materials applied in BSBs, such as quinones, phenothiazine, and zinc, commonly have a relatively low capacity[8,12,14]. The aqueous environment limits the application of some high-specific-energy electrodes (such as alkali anodes)[15]. These two fatal drawbacks result in a low energy density (Wh L$^{-1}$) of BSBs and constrain their wide application.

Herein, we develop nonaqueous BSBs based on Li-S chemistry. Through the synergistic effect of LiNO$_3$ and lithium bis(trifluoromethanesulfonyl)imide (LiTFSI), two organic solvents with different polarity successfully undergo salting-out stratification and provide sufficient ionic communication between the top and bottom electrolyte phases (Fig. 1a). The active species, lithium polysulfides, are confined to the bottom phase during charge and discharge and could not shuttle to the anode side due to the extraction effect. A schematic representation of this battery concept is shown in Fig. 1b. This design dramatically increases the overall energy density of the battery while avoiding an expensive separator. Based on the ultrahigh specific capacity of sulfur and the ultralow potential of a lithium anode, a high volume-energy density of 88.5 Wh L$^{-1}$ is achieved. The phase separation behavior and the solvation structure change of Li$^+$ during cycling are systematically studied by molecular dynamics (MD) simulations. In situ UV–Vis spectroscopy reveals that the active species (polysulfides) are localized in the bottom electrolyte phase during charge/discharge, so the self-discharge and shuttle effect of the Li-S BSB is significantly reduced. This proof-of-concept confirms the practicality of nonaqueous biphasic electrolyte systems and provides an idea to realize massive-scale energy storage with large capacitance.

## Results

### Design considerations of Li-S BSBs

Completely nonaqueous biphasic systems have been widely used in organic synthesis as an isolation technique[16,17]. They can be considered to be composed of a polar solvent and a low-polarity counter-solvent optimized to minimize the mutual solubility. However, directly

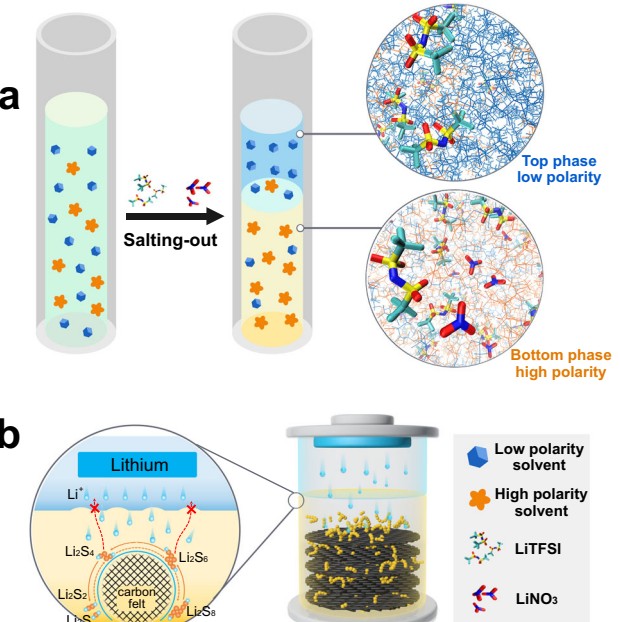

**Fig. 1 | Schematic illustration of Li-S BSBs. a** Stratification phenomenon between low polarity solvents and high polarity solvents salted-out by LiTFSI and LiNO$_3$. **b** Schematic representation of Li-S BSBs, in which LPSs are confined to the bottom phase during charge and discharge.

applying them as electrolyte systems to enable batteries to work properly is difficult. Solvents with low polarity cannot solvate conductive salts and conduct charge transfer efficiently. A self-stratified system can be applied in battery only when a delicate balance is achieved in the solubilities for the counter-solvent and electrolyte, and charge carriers travel freely between the two phases. In addition, the formed nonaqueous biphasic systems need to confine the cathode and anode active species in different phases and remain stable during battery cycling.

Based on the above considerations, we selected dimethylacetamide (DMA) and diethyl ether (DEE) as the mixed solvents due to the large difference in their dielectric constants ($\varepsilon_{DMA} = 37.78$, $\varepsilon_{DEE} = 4.33$)[18]. Two representative lithium salts, LiNO$_3$ and LiTFSI, separated the DMA and DEE hybrid solutions. A series of typical component screening experiments were performed (Supplementary Table 1), and their results are shown in Fig. 2a. We find that under the condition where the concentration of lithium ions in DMA is maintained at 2 M, more than twice the volume of DEE can induce a self-stratified system. Classical MD simulations under different ratios were also conducted, the final state was captured, and snapshots are displayed in Fig. 2b and Supplementary Fig. 1. Similar results show that apparent phase splitting behavior occurred in the No. 2, No. 3, and No. 7 systems. Supplementary Fig. 2 displays the DMA-DEE radial distribution function (RDF) at the beginning and end of MD simulations. A disparity can be found in the No. 2, No. 3, and No. 7 systems, in which g(r) significantly decreased at the end of the simulation, while the other systems did not exhibit distinct changes. Supplementary Fig. 3 shows the spatial density analysis of DEE and DMA from the No. 1 to No. 8 systems, in which DEE and DMA in No. 2, No. 3, and No. 7 have uneven distributions, and their distribution trends are opposite. The above simulation results indicate that phase splitting behavior occurs at the micro-level in the No. 2, No. 3, and No. 7 systems, which agrees with the experimental results.

Adequate Li$^+$ conductivity of both the top and bottom phases is a precondition for this biphasic self-stratified system to enable battery operation. Figure 2c exhibits the mean square displacement (MSD) of Li$^+$ in the No. 2, No. 3, and No. 7 systems. The slope of the No. 2 system is steeper than that of No. 2 and No. 7, implying faster diffusion of Li$^+$ in No. 2, and the corresponding diffusivity was estimated to be $1.33 \times 10^{-6}$ cm$^2$ s$^{-1}$. The Li$^+$ conductivities of the top and bottom phases in No. 2, No. 3, and No. 7 were also tested (Supplementary Fig. 4), and their results are summarized in Fig. 2d. The bottom phase in all three systems has high Li$^+$ conductivity. However, only No. 2 maintains the Li$^+$ conductivity in the top phase at the same order of magnitude as that in the bottom phase ($10^{-3}$ S cm$^{-1}$). Based on the above simulation and experimental results, the No. 2 biphasic system was finally selected for further research.

$^7$Li NMR, $^{19}$F NMR, and $^1$H NMR were conducted to accurately quantify the contents of each component in the top and bottom phases (Supplementary Fig. 5 and Supplementary Table 2; the calculation details can be found in the Supplemental Information). The results are summarized in Fig. 2e. The bottom phase is a DMA-rich phase, in which enriched 88.1% of total DMA. In contrast, the top phase is a DEE-rich phase. Most lithium salts gather in the bottom phase. In particular, there is almost no NO$_3^-$ in the top phase, while approximately 11.2% LiTFSI is in the top electrolyte. These results indicate that LiNO$_3$ dominated the salting-out phenomenon. When LiNO$_3$ is added into the DEE/DMA hybrid system, DMA molecules tend to interact with LiNO$_3$, and extrusion of DEE molecules out of the DMA solvent shell is induced. In contrast, organic lithium salt (LiTFSI) has a high solubility in low-polar solvents[19,20]. Therefore, the salting-out ability of LiTFSI is weak, which corresponds to the results for the previous No. 6 system in that LiTFSI cannot stratify DMA and DEE. The small amount of LiTFSI remaining in the top phase takes on the role of conducting Li$^+$[21].

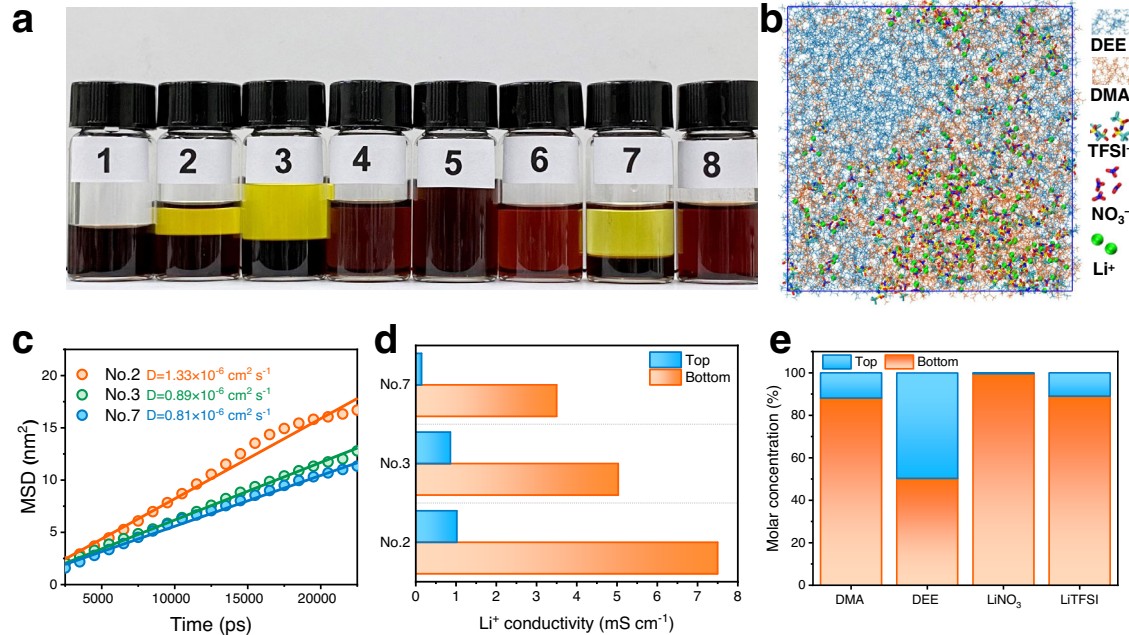

**Fig. 2 | Design considerations of Li-S BSBs. a** Phase separation of DMA-DEE systems with different amounts of LiTFSI and LiNO₃. The color is from the pigment dissolved in them. **b** Snapshot of the MD simulation of No. 2 system showing apparent phase separation behavior. **c** Calculated MSD of Li⁺ in the three biphasic systems (No. 2, No. 3, No. 7) as a function of the simulation time. The diffusion coefficient of Li⁺ was deduced by fitting. **d** Li⁺ conductivity of the top and bottom phases in No. 2, No. 3, and No. 7 biphasic systems. **e** Proportions of DEE, DMA, LiNO₃ and LiTFSI in the top and bottom phases of the No. 2 biphasic system.

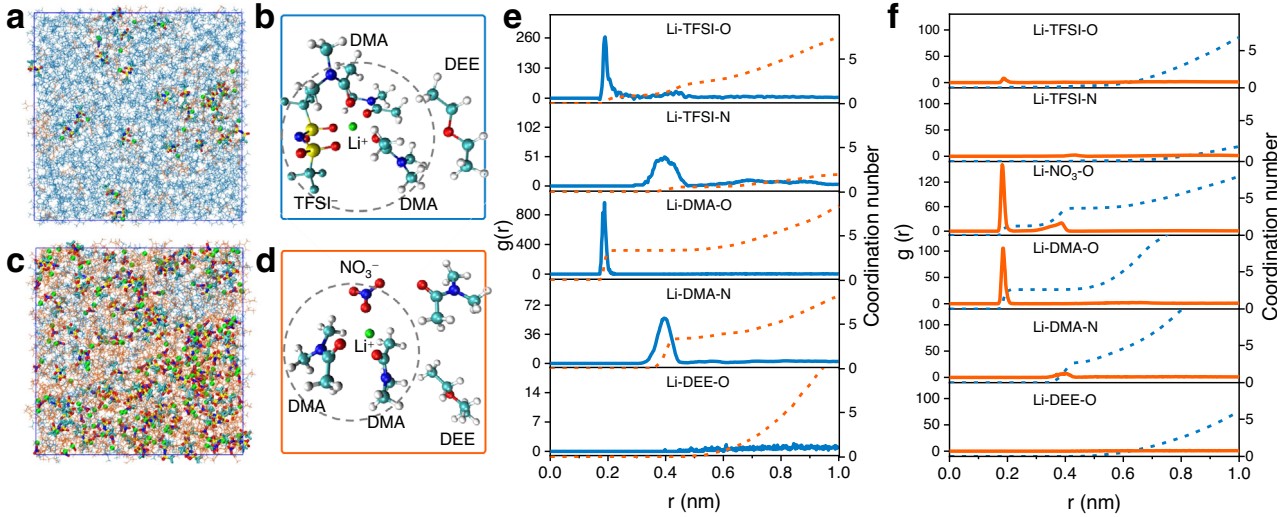

**Fig. 3 | Solvation state of Li⁺ in the DMA-DEE biphasic system. a, c** Snapshots of the MD simulation of the top phase (**a**) and bottom phase (**c**). **b, d** Representative Li⁺ solvation structure in the top phase (**b**) and bottom phase (**d**) extracted from the cMD simulations. **e** Li⁺ intermolecular RDFs in the top phase, including Li⁺/O (TFSI⁻), Li⁺/N (TFSI⁻), Li⁺/O (DMA), Li⁺/N (DMA), and Li⁺/O (DEE). **f** Li⁺ intermolecular RDFs in the bottom phase, including Li⁺/O (TFSI⁻), Li⁺/N (TFSI⁻), Li⁺/O (NO3⁻), Li⁺/O (DMA), Li⁺/N (DMA), and Li⁺/O (DEE).

## Electrolyte solvation structure and mass transfer process of Li⁺ in DMA-DEE systems

Free movement of Li⁺ in this DMA-DEE self-stratified system is a precondition for battery regular operation. Given the biphasic structure of BSBs with two different electrolyte solvation structures, the transfer of Li⁺ from one phase to another must accompany transformation of the solvation structure and energy. Based on the quantitative NMR results, MD simulations were conducted to evaluate the Li⁺ solvation structure in both the top and bottom phase electrolytes. The resulting data are shown in Fig. 3a, c. Analysis of the RDF data revealed that Li⁺ in the top and bottom phases displays a characteristic contact-ion pair structure

balanced between ion/solvent and anion/cation binding. DEE cannot interact with Li⁺ directly in the top phase, and the coordination number of DMA remains at approximately 3.3. TFSI⁻ exists in the primary solvation shell by directly interacting with Li⁺, and its coordination number to Li⁺ is near 1.0 (Fig. 3e). In the bottom phase (Fig. 3f), the Li⁺ coordination is dominated by DMA and NO₃⁻, with an average coordination number of 2.4 DMA and 1.3 NO₃⁻ per Li⁺. TFSI⁻ and DEE cannot exist in the first solvent shell but are still partially oriented by the influence of Li⁺. According to the RDF results, the solvation structure of Li⁺ in the top and bottom electrolyte phases was finally determined. Figure 3b, d demonstrate the most likely solvation structure for the top

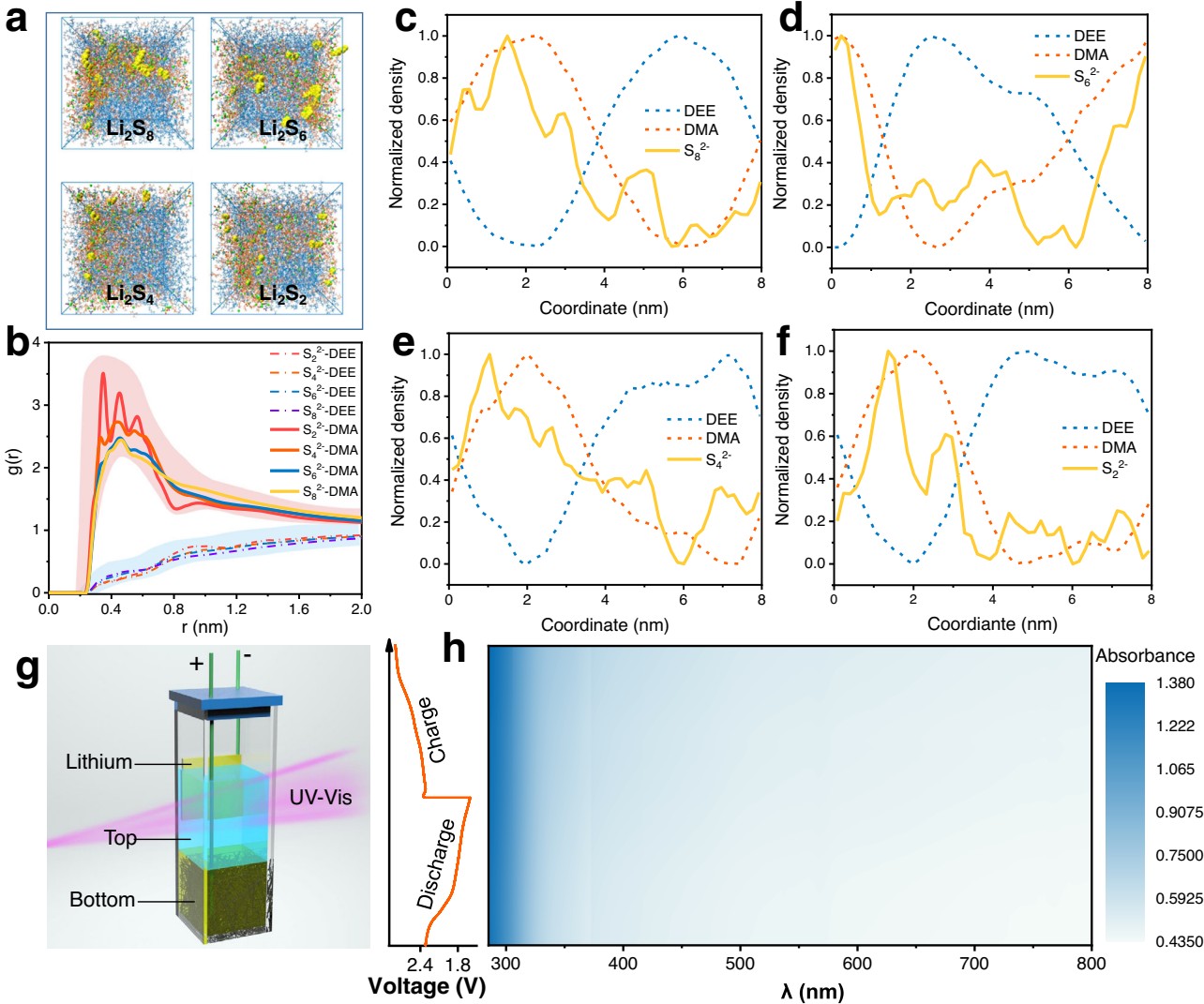

**Fig. 4 | Polysulfide-confinement effect of the DMA-DEE biphasic system.** **a** Snapshots of the MD simulation of the No. 2 biphasic system contains different intermediates ($Li_2S_8$, $Li_2S_6$, $Li_2S_4$, $Li_2S_2$). **b** RDFs of polysulfide ions between DEE and DMA. **c**, **d**, **e**, **f** Normalized spatial density distribution of DEE and DMA in No.2 biphasic system containing various polysulfides: (**c**) $Li_2S_8$, (**d**) $Li_2S_6$, (**e**) $Li_2S_4$, (**f**) $Li_2S_2$, **g** Schematic illustration of the cuvette battery. **h** 2D contour plot of the in situ UV/Vis spectrum (right side) and corresponding charge/discharge curve of the cuvette battery.

and bottom phases extracted from the MD simulation results. Density functional theory (DFT) was applied to determine the Gibbs energy of $Li^+$ transfer between the two phases. The solvation energy in the top $\Delta G_{sv}(Li^+, T)$ and bottom $\Delta G_{sv}(Li^+, B)$ phases is −273.475 and −232.425 kJ mol$^{-1}$ (Supplementary Fig. 6), respectively. According to the following Eq[22]:

$$\Delta G_t\left(Li^+, T \rightarrow B\right) = \Delta G_{sv}\left(Li^+, B\right) - \Delta G_{sv}\left(Li^+, T\right) \quad (1)$$

the Gibbs energy of transfer of $Li^+$ from the top phase to the bottom phase is 41.05 kJ mol$^{-1}$, which means that $Li^+$ has difficulty spontaneously migrating from the top phase to the bottom phase. This electrolyte system will effectively inhibit battery self-discharge if the anode is in the top phase while the cathode is in the bottom phase.

**Polysulfide-confinement effect of the DMA-DEE biphasic system**
A prerequisite for regular operation of a Li-S BSB is to ensure that active species cannot contact each other. In this DMA-DEE system, sulfur and its subsequent reduction products (lithium polysulfides, LPSs) were chosen as cathode materials due to their dissolution feature in DEE. As shown in Supplementary Fig. 7, $Li_2S_8$, $Li_2S_6$, and $Li_2S_4$ are

all dissolved in DMA, and the solutions turn brownish-red, while they cannot dissolve in DEE. MD simulations were first conducted to verify that this DMA-DEE system can constrain intermediates ($Li_2S_8$, $Li_2S_6$, $Li_2S_4$, and $Li_2S_2$) to a particular phase during battery charge/discharge. A certain amount of different LPSs was placed in the No. 2 system box, and their distribution was studied (Fig. 4a). Figure 4b summarizes the RDFs of polysulfide ions between DEE and DMA, in which most of the molecules around LPSs are DMA, whereas few are DEE. Figure 4c–f are the normalized spatial density distributions of DEE, DMA, and polysulfide ions in the four boxes in Fig. 4a. Polysulfide ions and DMA show similar distribution trends, while DEE shows the opposite trend. The above results demonstrate that the polysulfide ions are enriched in the DMA-rich phase while not diffusing into the DEE-rich phase. A certain amount of LPSs ($Li_2S_8$, $Li_2S_6$, and $Li_2S_4$, 10 mM) was added to this DMA-DEE electrolyte system, in which it can be found that the bottom phase is opaque due to collect almost all LPSs. However, the top phase exhibits a slight blue color, which is mainly due to the decomposition of LPSs into $S_3 \cdot$ radical in high donor number solvents and a small amount $S_3 \cdot$ radicals shuttle to the top phase[23,24]. (Supplementary Fig. 8) To eliminate this, 2,2,6,6-tetramethylpiperidine-1-oxy (TEMPO), a commonly peroxy radical scavenger[25,26], was applied to

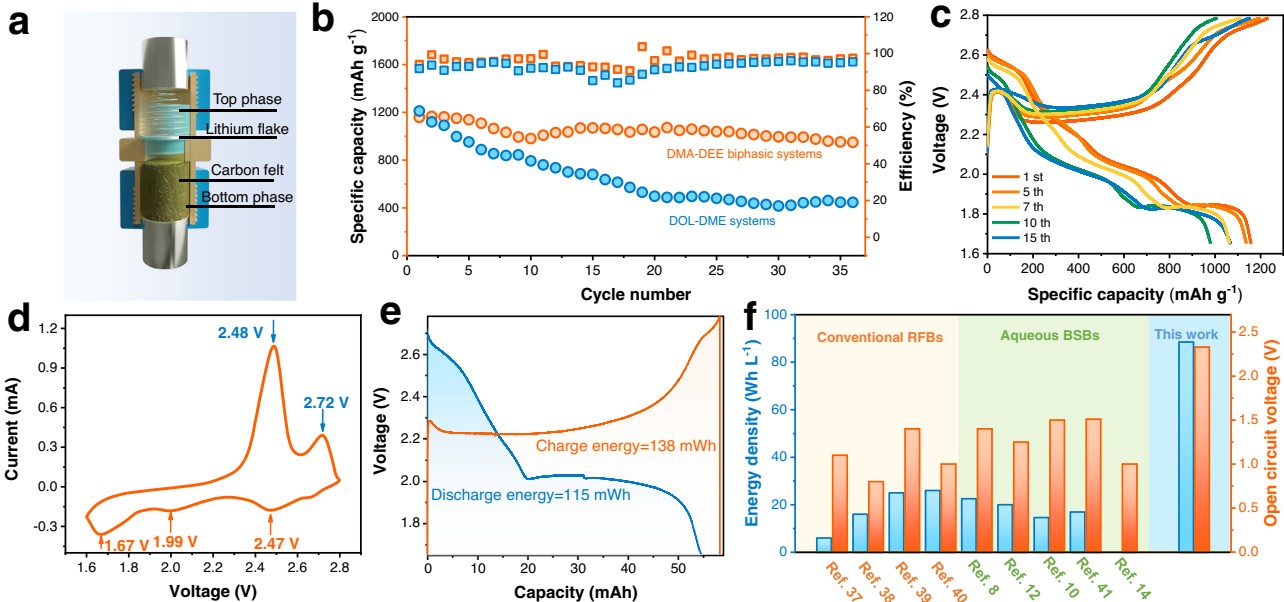

**Fig. 5 | Electrochemical performance of Li-S BSBs. a** Schematic illustration of Li-S BSBs. **b** Cycling performance of Li-S batteries using biphasic electrolyte system and conventional electrolyte. **c** Charge/discharge profiles of a Li-S BSB. **d** CV curves of a Li-S BSB at a scanning rate of 0.1 mV s⁻¹. **e** Capacity performance of the Li-S BSB with a high polysulfide content. The charge and discharge energies are 138 mWh and 115 mWh, respectively. **f** Comparison of the energy densities and supply voltage of some recently reported RFBs or BSBs with those in this work.

scavenging this little part radical due to its higher oxidizing ability than $S_3\cdot$ and relatively stable to the lithium anode[27]. A small amount of 2,2,6,6-tetramethylpiperidine-1-oxy (TEMPO, 0.5 mM) was added to the DMA-DEE biphasic electrolyte, thus successfully avoiding the shuttle of a few $S_3\cdot$ radicals. As shown in Supplementary Fig. 9, LPSs are extracted into the bottom phase, and the top phase exhibits a transparent state. This biphasic electrolyte system can recover spontaneously even after being disturbed by a strong external force (Supplementary Fig. 10 and Supplementary Movie 1), indicating that this biphasic self-stratified system may well constrain LPSs in the bottom phase during the RFB cycle. In situ UV-Vis spectrometry was conducted to directly verify the constraining properties of this biphasic system for LPSs. A particular cuvette battery was assembled to realize the real-time monitoring of the concentration of LPS in the top phase during battery cycling (Fig. 4g). The UV-Vis spectra of DMA solutions with different LPSs show an absorption peak between 360 and 400 nm, and the peak intensity decreases with the reduction of the $n$ in Li$_2$S$_n$ (Supplementary Fig. 11). In addition, the obvious signals that appear at 620 nm can be ascribed to the $S_3\cdot$ radical due to the LPSs dissolved in the high-donor-number solvent[23,24]. In comparison, the LPSs dissolved in DEE demonstrate no obvious absorption signal. As shown in Fig. 4h, the cuvette battery does not exhibit any absorption signal during either the first charge or discharge, which powerfully demonstrates that the LPSs formed during the cycling are well-constrained in the bottom phase and cannot shuttle to the top phase. Ex situ Raman tests were conducted to further illustrate the confined ability of LPSs in the biphasic electrolytes. The top phase electrolytes were sampled at various discharge/charge stages (Figure S12a). Their Raman spectra demonstrated a similar trend to the bare top electrolyte, and no characteristic peaks were found between 100–500 cm⁻¹ (Figure S12b), indicating no LPS shuttle to the top phase[28], and this is consistent well with the in situ UV-Vis spectrometry results.

Metallic lithium is generally unstable in solvents with high electronic constants, such as water, DMSO, and DMA. Therefore, expensive Li-conductive ceramic separators are commonly used in the reported works to enable lithium to be free from corrosion[29,30]. Interestingly, a steady state was maintained in our nonaqueous biphasic self-stratified system when lithium was in the top phase. As demonstrated in

Supplementary Fig. 13, the lithium immersed in DMA started to react after 5 h, and bubbles appeared on the lithium surface. In contrast, the lithium remained stable in the top phase after 15 days, while that in pure DMA was completely rusted. These results suggest that this DMA-DEE self-stratified system can allow the lithium metal battery to operate stably, avoiding the use of an expensive Li-conductive separator.

## Electrochemical performance of Li-S BSBs

A well-designed Swagelok battery was constructed to prove the concept of this Li-S BSB and evaluate the practicability of this unique system. As depicted in Fig. 5a and Supplementary Fig. 14, a coiled carbon felt loaded with sulfur was placed in the lower chamber, and soaked in the bottom electrolyte phase, and a piece of lithium flake was fixed in the top electrolyte phase by a spring. The phase interface is between the lithium flake and carbon felt. The redox behavior was first studied by cyclic voltammetry (CV). Three anodic peaks at 2.47 V, 1.99 V, and 1.67 V were detected during the anodic scan that can be ascribed to $S_8 \rightarrow S^{2-}_8/S^{2-}_6$, $S^{2-}_6 \rightarrow S^{2-}_4/S_3\cdot/S^{2-}_3$, and the final reduction to Li$_2$S, respectively[31]. The two obvious cathodic peaks at 2.48 V and 2.72 V are due to oxidation of Li$_2$S to sulfur or high-order LPSs (Fig. 5d).

Figure 5b shows the cycling performance of Li-S batteries with DMA-DEE biphasic systems and a traditional 1,3-dioxolane (DOL)−1,2-dimethoxyethane (DME) electrolyte, where the sulfur loading in carbon felt is approximately 2 mg and under a current density of 0.2 C (1 C = 1675 mAh g⁻¹). For the traditional electrolytes, the battery capacity dramatically declines in the first five cycles, and a value of only approximately 400 mAh g⁻¹ can be obtained after 30 cycles. This is mainly caused by the shuttle of LPSs in traditional electrolytes and their persistent corrosion on lithium metal anode. In contrast, the initial discharge capacity of Li-S BSBs is up to 1158 mAh g⁻¹, and the capacity remains above 1000 mAh g⁻¹ after 30 cycles. In addition, this DMA-DEE biphasic electrolyte system also endows Li-S BSBs with a higher coulombic efficiency than traditional electrolytes. Figure 5c displays the charge/discharge curves of Li-S BSBs. Three plateaus occur during discharge that correspond to the CV results. In general, the dielectric constant of the solvent and the plateau value demonstrate a negative correlation[32]. And as mentioned above, Li⁺ also needs

to overcome certain energy barriers when crossing the phase boundary during discharge. Therefore, the potential value for the last plateau is relatively lower compared to that in DOL/DME (around 2 V). Supplementary Fig. 15 shows the rate performance of Li-S BSB, in which the Li-S BSB delivers a specific discharge capacity of 783.3 mAh g$^{-1}$ even at a high current density of 1 C. The above cycling and rate performance verifies the feasibility of Li-S BSBs. The lithium anode in Li-S BSBs may affect the battery system's overall performance because there is almost no external pressure on it during battery operation, leading to dendrites growth and electrode surface deterioration. Electrochemical impedance spectroscopy (EIS) analyses were performed on Li-S BSBs at different cycle numbers, as shown in Supplementary Fig. 16. The impedance of lithium/electrolyte interphase (R3) increases rapidly as the battery continues to charge/discharge (from 9.57 to 68.97 Ω), indicating that the dendrite growth leads to an overgeneration of solid electrolyte interface (SEI)[33,34]. Supplementary Fig. 17 shows the surface morphology of lithium anode in Li-S BSBs at different cycle numbers, which corresponds well with EIS results.

Compared with common aqueous RFBs and BSBs, the biggest advantage of nonaqueous BSBs lies in their wider electrochemical window. In addition, DMA can dissolve more LPSs than DOL and DME[23,35,36]. Therefore, this Li-S BSB can achieve a higher energy density. A Li-S BSB with a high sulfur content (50 mg) was assembled in which contained 1.3 mL electrolyte, and its output energy is 115 mWh under 0.1 C (Fig. 5e). The volume energy density is up to 88.5 Wh L$^{-1}$, and the energy efficiency is 83.3%, which is higher than that of traditional RFBs[37-40] and existing BSBs[8,10,12,14,41] (Fig. 5f), demonstrating promising application prospects. Another prototype of Li-S BSBs with a non-static biphasic electrolyte was also fabricated to verify the feasibility of operating in a dynamic state. As shown in Supplementary Fig. 18a, b, this stirred Li-S BSB demonstrated similar charge/discharge profiles. A specific capacity of 764 mAh g$^{-1}$ is achieved at a current density of 0.5 mA cm$^{-2}$. Compared with the static state, this specific capacity is slightly decreased, which is mainly due to that insoluble LPSs near the end of discharge being challenging to attach to the current collector. Supplementary Fig. 18c and Supplementary Movie 2 exhibited the phase interface stability of Li-S BSBs under stirred environment.

## Discussion

The Li-S BSB introduced in this paper has a unique and straightforward structure, which provides a proof-of-concept of a nonaqueous biphasic electrolyte system in the energy storage field. The dissolved redox species, LPSs, in biphasic electrolyte systems will never contact the lithium counter electrode due to the extraction effect. Therefore, the notorious "shuttle effect" is completely avoided. This Li-S BSB delivered an open-circuit voltage of 2.33 V with a high energy density of 88.5 Wh L$^{-1}$, which pushes the energy densities of RFBs and provides an idea to realize massive-scale energy storage with large capacitance. The critical issue is ensuring the ionic conductivity between the two phases, and the dissolved species can stay in one phase well. In addition to the DMA-DEE biphasic system, other nonaqueous biphasic electrolyte systems could be potentially developed and applied in the energy storage system based on this design consideration. For example, another biphasic electrolyte system formed by the ternary of dimethyl sulfoxide (DMSO), 1,3-dioxolane (DOL), and DEE was fabricated, demonstrating the universality of nonaqueous BSBs (Supplementary Figs. 19, 20, Supplementary Note, and Supplementary Movie 3). It is worth mentioning that there is still a significant untapped opportunity in the overall energy density of Li-S BSBs due to the active species in Li-S BSBs are far below their dissolution limits now. Current lithium metal anode technologies are difficult to operate stably in the case of huge area capacity, especially in the absence of external pressure to inhibit the arbitrary growth of dendrites[42,43]. Therefore, further research on lithium anode protection is essential to promote the better application of nonaqueous BSBs.

## Methods

### Materials

Anhydrous Ether, anhydrous dimethylacetamide were purchased from Sigma-Aldrich or Aladdin (Shanghai, China). Lithium sulfide and element sulfur were purchased from Alfa Aesar. TEMPO, carbon disulfide, and anhydrous dimethyl sulfoxide were purchased from Aladdin. LiTFSI, LiNO$_3$, and dioxolane were purchased from https://www.dodochem.com/. The carbon felt (thickness: 1 mm) was purchased from Beijing Jinglong Special Carbon Technology Corporation (Beijing, China). The carbon felt was ultrasonically washed twice before use with alternating alcohol and acetone, and other chemicals were used as received. The lithium polysulfides were synthesized by adding a certain mole ratio of lithium sulfide to element sulfur into different solvents under heat. The lithium flakes were obtained from China Energy Lithium Co., Ltd (diameter: 12 mm, thickness: 0.45 mm).

### Sulfur/carbon felt composite cathode

A certain amount of sulfur powders was dissolved in carbon disulfide (CS$_2$) and dropped on a specific size of carbon felt. After CS$_2$ volatilizes completely, the carbon felt was sealed in an ampoule with an argon environment and then transferred to an oven at 155 °C for 6 h under the Ar atmosphere. The ampoule bottle was broken to get the Sulfur/carbon felt composite cathode used in this manuscript.

### In-situ UV-Vis test

A coiled carbon felt (1.5 cm * 15 cm) loaded with 5 mg sulfur was placed in the bottom of the cuvette, then 3 ml electrolyte was added to it, keeping the phase interface slightly above the carbon blanket. A piece of lithium was immersed into the top phase near the inner wall. Two titanium wires were connected to carbon felt and lithium flake, respectively, the other side pierced the rubber plug, and the punctures were sealed with silicone rubber. Covering the bottom phase with black electrical tape allows spectrum through the top phase. The in-situ UV/Vis Spectroscopy was operated on a UV Lambda 750 UV/Vis/NIR spectrometer. This cuvette battery was discharged/discharged under a current density of 0.5 C. After taking the batteries' initial spectra, cycling was started using the LAND CT2001A battery test system. Meanwhile, UV/Vis spectra were recorded from 200 to 800 nm during the battery discharge/charge.

### Battery assembly

A well-designed Swagelok battery was used to test the electrochemical performance of Li-S BSBs. Its internal structure is wide at both ends and thin in the middle. A coiled carbon felt (0.5 × 14 cm$^2$) loaded with 2 mg sulfur was placed in the lower chamber, and a stainless-steel rod was applied to secure the carbon felt and act as a conductive electrode. Then 1.3 mL electrolytes were added to the Swagelok cell and kept the phase interface in the middle of the battery case. A piece of lithium flake was then fixed in the top electrolyte phase by a spring, and a stainless-steel rod was applied to squeeze the spring and act as a conductive electrode.

For Li-S BSBs with a stirring system, one piece of carbon felt (1 × 1 cm$^2$) was clamped with nickel foam and fixed to the electrode clip as the cathode current collector. Cutting the lithium flake into a size of 0.7 × 1.2 cm$^2$ and fixing it on the negative electrode clip. 12 mL electrolytes (4 mL DMA + 8 mL DEE + 276 mg LiNO$_3$ + 1150 mg LiTFSI) were added to a customized electrolytic cell, sinking the electrode into the electrolyte and adjusting the electrode height to keep the phase interface was located between these two electrodes. The speed of magneton was controlled at 300 rpm/min. All the batteries should remain vertical during assembly and subsequent testing to eliminate the possibility that the catholyte contact directly to lithium metal anode.

## Electrochemical measurements

All BSBs were placed in a thermostatic chamber (25 °C) and remain in a vertical position during the electrochemical test. The cyclic voltammetric and electrochemical impedance spectroscopy (EIS) were tested on the CHI710 (Shanghai Chenhua instrument corporation) electrochemical workstation, and the frequency ranges from 1000000 to 0.01 Hz. The BSBs were galvanostatically cycled between 1.65 and 2.79 V on the CT2001A cell test instrument (Wuhan LAND Electronic corporation). The specific energy densities (*E*) of the BSB was calculated by

$$E = \frac{1}{V} \int_0^t IU \, dt \qquad (2)$$

where *V* is the electrolyte volume, I and U refer to current and voltage, respectively. The ionic conductivity was tested using the 2032 coin cells, in which two stainless steel flakes were used as the plug electrode and separated by a glass fiber separator. The ionic conductivity, σ, was calculated using

$$\sigma = \frac{d}{RA} \qquad (3)$$

where d is the distance between two stainless-steel electrodes, A is the area of the plug electrode, and R is the resistance measure from EIS.

## NMR and Raman measurements

All NMR spectra were measured with a Bruker 600 MHz NMR spectrometer. For the $^{1}$H NMR, 500 μL electrolytes in top phase and bottom phase were added into the nuclear tube, respectively, and then each tube was filled with 100 μL [D6]-DMSO. For the $^{7}$Li NMR and $^{19}$F NMR, 600 μL electrolytes in the top and bottom phases were added into the nuclear tube, respectively, and then tested directly. Raman spectra of electrolytes were conducted with a Horiba Jobin Yvon HR Evolution Raman spectrometer under a Raman laser wavenumber of 633 nm.

## Classical molecular dynamics simulations

Molecular dynamics (MD) simulations were performed with Gromacs, version 2019.6[44]. The simulation system was constructed with PACKMOL software[45] by uniformly mixing different amounts of LiNO$_3$, LiTFSI, lithium polysulfides, DEE, DMA, DOL and DMSO molecules. The molecule number of each component were summarized in Supplementary Table 3. DEE, DMA, DOL, DMSO, NO$_3^-$, TFSI$^-$ and polysulfides were modeled by the GAFF force field[46], with bonded parameters for polysulfides obtained from the reported work[47], for Li$^+$ the parameter was identical to Wang et al.[48] RESP2(0.5) partial charges proposed by Schauperl et al.[49] were used for the species involved in this study, by first optimizing the molecular geometry at the B3LYP/TZVP level of theory in Gaussian 16, and then fitting to electrostatic potential calculated at the B3LYP/ma-TZVP level of theory with the help of Multiwfn program[50]. Van der Waals interactions were described by the Lennard Jones (LJ) potential truncated at 1.2 nm, with LJ interaction parameters between unlike atom pairs generated by Lorentz-Berthelot combining rules. Long range electrostatic interactions were calculated with the particle-mesh Ewald method, with a grid spacing of 0.12 nm and a pme-order of 4. Initial configurations were energy-minimized and equilibrated at 298.15 K and 1 bar. Subsequently, 20-ns production runs (50 ns for the system in Fig. 2 and S1) with a time step of 1 fs were carried out at 298.15 K and 1 bar with the Nose-Hoover thermostat and Parrinello-Rahman barostat. The leap-frog algorithm was used to integrate the Eqs of motion. Three-dimensional periodic boundary conditions were applied throughout simulations.

Cumulative distribution function (CDF, *n(r)*) between component A and component B were calculated though radial distribution function (RDF, *g(r)*) with the following Eq:

$$n_{AB}(r) = \int_0^r 4\pi r^2 \rho_B g_{AB}(r) \mathrm{d}r \qquad (4)$$

In which $\rho_B$ is the average number density of the B molecule in the electrolyte bulk phase, *r* is the distance between A and B. The coordinate numbers in the Li$^+$ first solvation shell were estimated from the integral of the first valley of the RDF.

The MSD of Li$^+$ at time *t*, and the corresponding diffusion coefficient *D* of Li$^+$ were calculated by

$$MSD(t) = \frac{1}{N_{Li^+}} \sum_{i=1}^{N} |r_i(t) - r_i(0)|^2 \qquad (5)$$

$$D_{Li^+} = \lim_{t \to \infty} \frac{MSD(t)}{6t} \qquad (6)$$

Where $N_{Li^+}$ is the total number of Li$^+$ in the electrolyte, $r_i(t)$ and $r_i(0)$ are the position at time t and the initial position of Li$^+$ in the simulation box, respectively.

## DFT computational methods

The DFT calculations were conducted in the Gaussian 16 suite of programs, and the rb3lyp density functional method was employed in this work. The 6–31 G* basis set was used for all the atoms in the geometry optimizations. Vibrational frequency analyses at the same level of the theory were performed on all optimized structures to characterize stationary points as local minima or transition states. Coordinates of structures used in this work can be obtained from Supporting Data.

## Data availability

The data that support the findings of this study are available within the article and its Supplementary Information files or from the corresponding author upon reasonable request. Source data are provided in this paper. Source data are provided with this paper.

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

## Acknowledgements

This project is supported from the National Natural Science Foundations of China (Grant No. 52071226, 51872193, and U21A20332), Natural Science Foundations of Jiangsu Province (Grant No. BK20181168 and BK20201171), Key R&D Project funded by Department of Science and Technology of Jiangsu Province (Grant No. BE2020003-3) awarded to C.Y. and T.Q. We also acknowledged the support from the Natural Science Foundation of the Jiangsu Higher Education Institutions of China (Grant No. 19KJA210004), the Priority Academic Program Development of Jiangsu Higher Education Institutions (PAPD).

## Author contributions

C.Y. and T.Q. proposed and supervised this project. Z.W., J.Z., and Y.Z. synthesized the samples and performed the characterizations and electrochemical measurements. H.J. conducted the theoretical simulations. J.L. offered help in the material characterizations. Z.W. wrote this paper, J.L., C.Y. and T.Q. revised the paper.

## Competing interests

The authors declare no competing interests.
