## [Peer Review File · Nature Communications]

Exploiting nonaqueous self-stratified electrolyte systems toward large-scale energy storageREVIEWER COMMENTS

Reviewer #1 (Remarks to the Author):

This manuscript reports on a biphasic membrane-less redox battery concept with the possibility of higher energy density. The basic idea is ions can partition in two separate phases, thus eliminating the need for a membrane. The concept is very much similar to the molten metal batteries (density differences) of the MIT type of technology. Also, there have been numerous other membrane-less concepts in fuel cells and batteries over the years, so the idea is not all that new. However, they selected a Li-S system in two immiscible organic solvents that seems to work with reasonable performance. This reviewer is not aware of similar studies on this type of system in the membrane-less configuration. And, they have done a nice job of studying the phases to understand the solvation and structure of the adjoining fluids. The science is certainly the strength of the paper and is thorough and well thought out.

I do have some specific comments regarding potential impact of the work. First, they chose a very specific system to study and the results are quite limited to this Li-S DMA/DEE redox system. Second, the energy density is 88 Wh/l as compared to typical aqueous RFBs of 0-40 Wh/l. This is not a substantial increase in energy density and does not make that much difference for a large stationary storage system. Third, a stationary prototype was demonstrated instead of a small prototype flow battery. Phase boundaries between two flowing immiscible phases are expected to have different interphase transport characteristics as compared to stationary interphases. So the experimental results of the prototype battery are not necessarily representative of what might be expected.

Reviewer #2 (Remarks to the Author):

In this paper, authors present a membraneless Li-S battery using a self-stratified electrolyte system. The composition of the presented battery is interesting, and various analyses had been conducted. However, in this research, authors compared their results with redox flow battery (RFB) results. However, the BSB suggested by authors is flowless battery, and there is no proof whether the two electrolytes are normally separated when actual flow exists. Therefore, it is inappropriate to compare the battery designed by the authors with RFB. If authors' final goal is flow battery design, battery test results showing flow behavior of electrolytes should be mainly presented. Therefore, the present format of manuscript is not ready to be well evaluated. In addition, there are additional comments as follows.

1. On pg.2, 'On the other hand, most electrode~~" part needs to be proved. References must be presented according to the sentence.
2. It is appropriate to present the principle and references in the section 'To limit this, a small amount of ~~~' on pg.6.
3. In Supplementary Figure 3, there is no numbering of each picture.
4. In BSB system of this study, upper and lower liquid layers exist. If these two solutions are mixed by external impacts, etc., additional explanation of the following questions can improve the novelty of this study: (i) Is it possible to separate electrolytes naturally again?, (ii) Is there any other permanent loss if it is separated?

Reviewer #3 (Remarks to the Author):

This is a very interesting paper which successfully extended the concept of self-stratified battery to Li-S chemistry and largely increased the energy density. I am very impressed that DMA and DEE form a biphasic system with lithium-ion conductivity in both phases and high polysulfide solubility in only one phase. The authors must have tried many recipes to get such good result. The conclusions are supported with both MD simulation results and experimental data. I strongly suggest to publish this paper. Here, I have some small suggestions to further improve the manuscript:

1. The evolution of the Li anode should be studied more extensively.

2. More EIS measurements may be helpful to explain the performance decaying mechanism.
3. In Fig. 5f, the open circuit voltage of this work was measured at the beginning of the discharge process. I think it is more instructive to exhibit the open circuit voltage at 50% SOC.

Reviewer #1:

This manuscript reports on a biphasic membrane-less redox battery concept with the possibility of higher energy density. The basic idea is ions can partition in two separate phases, thus eliminating the need for a membrane. The concept is very much similar to the molten metal batteries (density differences) of the MIT type of technology. Also, there have been numerous other membrane-less concepts in fuel cells and batteries over the years, so the idea is not all that new. However, they selected a Li-S system in two immiscible organic solvents that seems to work with reasonable performance. This reviewer is not aware of similar studies on this type of system in the membrane-less configuration. And, they have done a nice job of studying the phases to understand the solvation and structure of the adjoining fluids. The science is certainly the strength of the paper and is thorough and well thought out.

(1) I do have some specific comments regarding potential impact of the work. First, they chose a very specific system to study and the results are quite limited to this Li-S DMA/DEE redox system.

Response:

Thanks for the reviewer's careful and professional review. In the choice of redox conjugate pair applied in the nonaqueous biphasic electrolyte systems, the lithium anode has the lowest electrode potential and the highest theoretical specific capacity, and its combination with sulfur cathode holds promise for next-generation energy storage. Therefore, we selected Li-S chemistry to be applied in the nonaqueous biphasic electrolyte systems. The nonaqueous biphasic electrolyte systems were exploited based on the "salting-out" effect. In the previous version, DMA and DEE were studied as phase-splitting solvents due to the significant difference in their dielectric constants. In fact, based on the guidance of this methodology, some other stratified electrolyte systems with similar effects can also be developed. According to your valuable comment, we further developed another nonaqueous biphasic electrolyte system formed by the ternary mixture of dimethyl sulfoxide (DMSO), 1,3-dioxolane (DOL), and DEE, as shown in Figures S17 and S18, demonstrating the universality of this design consideration.

Page No. 9 in MS:

In addition to the DMA-DEE biphasic system, other nonaqueous biphasic electrolyte systems could be potentially developed and applied in the energy storage system based on this design consideration. For example, another biphasic electrolyte system formed by the ternary mixture of dimethyl sulfoxide

(DMSO), 1,3-dioxolane (DOL), and DEE was fabricated, demonstrating the universality of nonaqueous BSBs (Supplementary Figures 17 and 18, Supplementary Note, and Supplementary Movie 3).

Page No. 19 and 20 in SI:

Supplementary Figure 17. Design consideration of DMSO/DOL/DEE biphasic electrolyte system

a-b. Snapshots of the MD simulation results of DMSO/DEE system without/with DOL.

c. The normalized spatial density distribution of DMSO, DEE, and DOL in the simulation box.

d. Calculated MSD of Li^+ in DMSO/DEE system with/without DOL as a simulation time function. The diffusion coefficient of Li^+ was deduced by fitting.

e. Li^+ conductivity of the top and bottom phases in DMSO/DEE/DOL biphasic electrolyte system.

Supplementary Figure 18. Polysulfide-confinement of DMSO/DEE/DOL biphasic electrolyte system

a. Snapshots of the MD simulation results of DMSO/DEE/DOL system containing different lithium polysulfides.

b. RDFs of polysulfide ions between DMSO and DEE.

c-f. Normalized spatial density distribution of DMSO and DEE in DMSO/DEE/DOL biphasic system containing various polysulfides.

g. Spontaneous recovery process of DMSO/DEE/DOL biphasic electrolyte system after external disturbance. 10 mM of Li_2S_8 was dissolved in the electrolyte.

h. Charge/discharge profiles of a Li-S BSB used DMSO/DEE/DOL biphasic electrolyte system.

Supplementary note:

To demonstrate the universality of our design considerations for nonaqueous biphasic electrolyte systems, we further developed another biphasic electrolyte system formed by the ternary mixture of dimethyl sulfoxide (DMSO), 1,3-dioxolane (DOL), and DEE. DMSO has high permittivity

($\epsilon_{\text{DMSO}}=48.9$) and high LPSs solubility, enabling it to spontaneously separate from DEE under the salting-out effect of lithium salt and constrain LPSs effectively (Supplementary Figure 17). The addition of DOL can significantly enhance the Li^+ conductivity between the bottom and top phases. As demonstrated in Supplementary Figure 17c, the diffusivity of DMSO/DEE system with DOL was estimated to be $1.374 \times 10^{-6} \text{ cm}^2 \text{ s}^{-1}$, almost twice as much as without DOL. Finally, the Li^+ conductivity of the bottom/top phase in DMSO/DEE/DOL biphasic electrolyte system reaches 5.2 and 0.48 mS cm^{-1} , respectively.

As demonstrated in Supplementary Figure 18, this DMSO/DEE/DOL biphasic electrolyte system can also effectively confine different LPS species in the bottom phase and resist external interference. The Li-S BSB based on this biphasic electrolyte system was also fabricated, which delivers a specific discharge capacity of 815 mAh g^{-1} , demonstrating its potential in nonaqueous BSBs.

(2) Second, the energy density is 88 Wh/l as compared to typical aqueous RFBs of 0-40 Wh/l. This is not a substantial increase in energy density and does not make that much difference for a large stationary storage system.

Response:

Thanks for the reviewer's professional review. The energy density of 88.5 Wh/L demonstrated by Li-S BSBs in this manuscript was not very impressive compared to conventional RFBs. However, 50 Wh/L is almost the limit of the energy density of conventional RFBs, such as all vanadium batteries, where the concentrations of active species are generally above 1 M. It is worth mentioning that the active species applied in our Li-S BSBs are far below their limiting concentrations (the concentration of Li_2S_8 is only about 0.15 M). Therefore, there is a significant untapped opportunity in the overall energy density of Li-S BSBs. The main obstacle for further enhancing energy density is likely to come from the anode side. The lithium electrode must tolerate the repeated high area capacity Li^+ plating/stripping, which is still a serious challenge. Considering the capacity and stability of the lithium anode, we constructed the prototype of Li-S BSBs with sulfur content of only around 50 mg, and conducted a preliminary conceptual verification of their performance, achieving an energy density of 88.5 Wh/L. Future research on robust lithium anode and battery prototype optimization is expected to improve the overall energy density further. Thanks for your valuable comment again, and we have added the above discussion in the Discussion section of the revised manuscript.

Page No. 9 in MS:

It is worth mentioning that there is still a significant untapped opportunity in the overall energy density of Li-S BSBs due to the active species in Li-S BSBs are far below their dissolution limits now. Current lithium metal anode technologies are challenging to operate stably in the case of huge area capacity, especially in the absence of external pressure to inhibit the arbitrary growth of dendrites.^{42,43} Therefore, further research on lithium anode protection is essential to promote the better application of nonaqueous BSBs.

Page No. 17 in MS:

42. Weber, R. *et al.* Long cycle life and dendrite-free lithium morphology in anode-free lithium pouch cells enabled by a dual-salt liquid electrolyte. *Nat. Energy* **4**, 683–689 (2019).
43. Zhou, J. *et al.* Healable Lithium Alloy Anode with Ultrahigh Capacity. *Nano Lett.* **21**, 5021–5027 (2021).

(3) Third, a stationary prototype was demonstrated instead of a small prototype flow battery. Phase boundaries between two flowing immiscible phases are expected to have different interphase transport characteristics as compared to stationary interphases. So the experimental results of the prototype battery are not necessarily representative of what might be expected.

Response:

Regarding your concern about the characteristics of the phase interphase in a dynamic environment, we acknowledge that investigating the battery properties under dynamic conditions is significant because a dynamic electrolyte is beneficial to enhance the electrode kinetics and reduce the area of the current collector. Therefore, we introduced a stirring system into Li-S BSBs to create a non-static phase boundary and preliminarily verified the electrochemical performance of BSBs under stirred environment (Figure S16). We hope this strategy is acceptable, and thanks again for your valuable advice.

Page No. 8 in MS:

Another prototype of Li-S BSBs with a non-static biphasic electrolyte was also fabricated to verify

the feasibility of operating in a dynamic state. As shown in Supplementary Figures 16a and 16b, this stirred Li-S BSB demonstrated similar charge/discharge profiles. A specific capacity of 764 mAh g⁻¹ is achieved at a current density of 0.5 mA cm⁻². Compared with the static state, this specific capacity is slightly decreased, which is mainly due to that the insoluble LPSs near the end of discharge is challenging to attach to the current collector. Supplementary Figure 16c and Supplementary Movie 2 exhibited the phase interface stability of Li-S BSBs under stirred environment.

Page No. 11 in MS:

For Li-S BSBs with a stirring system, one piece of carbon felt (1×1 cm²) was clamped with nickel foam and fixed to the electrode clip as the cathode current collector. Cutting the lithium flake into a size of 0.7×1.2 cm² and fixing it on the negative electrode clip. 12 mL electrolytes (4 mL DMA + 8 mL DEE + 276 mg LiNO₃ + 1150 mg LiTFSI) were added to a customized electrolytic cell, sinking the electrode into the electrolyte and adjusting the electrode height to keep the phase interface was located between these two electrodes. The speed of magneton was controlled at 300 rpm/min.

Page No. 18 in SI:

Supplementary Figure 16. a. Schematic illustration of the Li-S BSB with the stirring system. **b.** Charge/discharge profiles of Li-S BSBs under stirred environment. **c.** Phase interface stability of Li-S BSB during charge and discharge under stirred environment.

Reviewer #2:

In this paper, authors present a membraneless Li-S battery using a self-stratified electrolyte system. The composition of the presented battery is interesting, and various analyses had been conducted. However, in this research, authors compared their results with redox flow battery (RFB) results. However, the BSB suggested by authors is flowless battery, and there is no proof whether the two electrolytes are normally separated when actual flow exists. Therefore, it is inappropriate to compare the battery designed by the authors with RFB. If authors' final goal is flow battery design, battery test results showing flow behavior of electrolytes should be mainly presented. Therefore, the present format of manuscript is not ready to be well evaluated.

Response:

Thanks for the reviewer's careful review and valuable comments. In the previous version, we constructed a stationary prototype of Li-S BSBs, in which the cathode current collector, carbon felt, fills almost the entire bottom phase, thus ensuring a sufficient reaction degree of active species. But we acknowledge that dynamic electrolyte is beneficial to enhance the electrode kinetics and reduce the area of the current collector, thus is expected to optimize the battery structure further. Therefore, according to your professional comments, we introduced a stirring system into Li-S BSBs to create a non-static phase boundary, which verified the stability of the top/bottom phase interface under dynamic conditions and the feasibility of BSBs when actual flow exists (Figure S16). In addition, the design intention of BSBs was to provide additional options for large-scale energy storage due to their simplified battery architecture compared to traditional flow batteries. Therefore, we think it is reasonable to compare our results with conventional RFBs and reported BSBs.

Page No. 8 in MS:

Another prototype of Li-S BSBs with a non-static biphasic electrolyte was also fabricated to verify the feasibility of operating in a dynamic state. As shown in Supplementary Figures 16a and 16b, this stirred Li-S BSB demonstrated similar charge/discharge profiles. A specific capacity of 764 mAh g^{-1} is achieved at a current density of 0.5 mA cm^{-2} . Compared with the static state, this specific capacity is slightly decreased, which is mainly due to that insoluble LPSs near the end of discharge being challenging to attach to the current collector. Supplementary Figure 16c and Supplementary Movie 2 exhibited the phase interface stability of Li-S BSBs under stirred environment.

Page No. 11 in MS:

For Li-S BSBs with a stirring system, one piece of carbon felt ($1 \times 1 \text{ cm}^2$) was clamped with nickel foam and fixed to the electrode clip as the cathode current collector. Cutting the lithium flake into a size of $0.7 \times 1.2 \text{ cm}^2$ and fixing it on the negative electrode clip. 12 mL electrolytes (4 mL DMA + 8 mL DEE + 276 mg LiNO_3 + 1150 mg LiTFSI) were added to a customized electrolytic cell, sinking the electrode into the electrolyte and adjusting the electrode height to keep the phase interface was located between these two electrodes. The speed of magneton was controlled at 300 rpm/min.

Page No. 18 in MS:

Supplementary Figure 16. a. Schematic illustration of the Li-S BSB with the stirring system. **b.** Charge/discharge profiles of Li-S BSBs under stirred environment. **c.** Phase interface stability of Li-S BSB during charge and discharge under stirred environment.

In addition, there are additional comments as follows.

1. On pg.2, 'On the other hand, most electrode~~" part needs to be proved. References must be presented according to the sentence.

Response:

Thanks for the reviewer's careful review. Related references have been presented in the revised manuscript.

Page No.2 in MS:

On the other hand, most electrode materials applied in BSBs, such as quinones, phenothiazine, and zinc, commonly have a relatively low capacity.^{8,12,14}

Page No.2 in MS:

8. Navalpotro, P., Palma, J., Anderson, M. & Marcilla, R. A Membrane-Free Redox Flow Battery with Two Immiscible Redox Electrolytes. *Angew. Chem.* **129**, 12634–12639 (2017).
12. Meng, J. *et al.* A Stirred Self-Stratified Battery for Large-Scale Energy Storage. *Joule* **4**, 953–966 (2020).
14. Navalpotro, P. *et al.* Exploring the Versatility of Membrane-Free Battery Concept Using Different Combinations of Immiscible Redox Electrolytes. *ACS Appl. Mater. Interfaces* **10**, 41246–41256 (2018).

2. It is appropriate to present the principle and references in the section 'To limit this, a small amount of ~~' on pg.6.

Response:

Thanks for the reviewer's professional review and valuable comment. A brief potential principle and references of TEMPO scavenging radicals have been presented in this revised manuscript.

Page No.6 in MS:

To eliminate this, 2,2,6,6-tetramethylpiperidine-1-oxy (TEMPO), a commonly peroxy radical scavenger,^{25,26} was applied to scavenging this little part radical due to its higher oxidizing ability than S₃· and

relatively stable to the lithium anode.²⁷ A small amount of 2,2,6,6-tetramethylpiperidine-1-oxyl (TEMPO, 0.5 mM) was added to the DMA-DEE biphasic electrolyte, thus successfully avoiding the shuttle of a few S₃[·] radicals.

Page No.6 in MS:

25. Tkacheva, A. *et al.* TEMPO-Ionic Liquids as Redox Mediators and Solvents for Li-O₂ Batteries. *J. Phys. Chem. C* **124**, 5087–5092 (2020).
26. Barton, D. H. R., Le Gloahec, V. N. & Smith, J. Study of a new reaction: Trapping of peroxy radicals by TEMPO. *Tetrahedron Lett.* **39**, 7483–7486 (1998).
27. Chen, J. *et al.* Selection of Redox Mediators for Reactivating Dead Li in Lithium Metal Batteries. *Adv. Energy Mater.* (2022) doi:10.1002/aenm.202201800.

3. In Supplementary Figure 3, there is no numbering of each picture.

Response:

Thanks for the reviewer's careful review. The relevant pictures have been corrected in this revised manuscript.

Supplementary Figure 3. The normalized spatial density distribution of DEE and DMA in different systems.

4. In BSB system of this study, upper and lower liquid layers exist. If these two solutions are mixed by external impacts, etc., additional explanation of the following questions can improve the novelty of this study: (i) Is it possible to separate electrolytes naturally again?, (ii). Is there any other permanent loss if it is separated?

Response:

Thanks for the reviewer's professional review and valuable suggestions.

For question (i), we conducted external interference experiments on the biphasic electrolyte systems, demonstrating that the biphasic electrolytes could recover spontaneously after mixing with external force.

This biphasic electrolyte system can recover spontaneously even after being disturbed by a strong external force (Supplementary Figure 10 and Supplementary Movie 1), indicating that this biphasic self-stratified system may well constrain LPSs in the bottom phase during the RFB cycle.

Page No. 12 in SI:

Supplementary Figure 10. The spontaneous recovery process of biphasic electrolyte system after external disturbance. 10 mM of Li_2S_8 was dissolved in the electrolyte.

For question (ii), drastic external interference is obviously adverse because the catholyte is possible to contact directly with the lithium anode, resulting in a depletion of metallic lithium and reduction of the overall stability for energy storage systems. Therefore, Li-S BSBs need a relatively stable operation environment, which is also why they are suitable for large-scale energy storage applications. We have added this explanation in this revised manuscript, and thanks for your suggestions again.

Page No. 11 in MS:

All the batteries should remain vertical during assembly and subsequent testing to eliminate the possibility that the catholyte comes directly into contact with lithium metal anode.

Reviewer #3 (Remarks to the Author):

This is a very interesting paper which successfully extended the concept of self-stratified battery to Li-S chemistry and largely increased the energy density. I am very impressed that DMA and DEE form a biphasic system with lithium-ion conductivity in both phases and high polysulfide solubility in only one phase. The authors must have tried many recipes to get such good result. The conclusions are supported with both MD simulation results and experimental data. I strongly suggest to publish this paper. Here, I have some small suggestions to further improve the manuscript:

1. The evolution of the Li anode should be studied more extensively.

Response:

Thanks for the reviewer's positive comments and valuable suggestions. The evolution of lithium anode has been studied by SEM and EIS in this revised manuscript.

Page No. 8 in MS:

The lithium anode in Li-S BSBs may affect the battery system's overall performance because there is almost no external pressure on it during battery operation, leading to dendrites growth and electrode surface deterioration. Electrochemical impedance spectroscopy (EIS) analyses were performed on Li-S BSBs at different cycle numbers, as shown in Supplementary Figure 14. The impedance of lithium/electrolyte interphase (R_3) increases rapidly as the battery continues to charge/discharge (from 9.57 to 68.97 Ω), indicating that the dendrite growth leads to an overgeneration of solid electrolyte interface (SEI).^{32,33} Supplementary Figure 15 shows the surface morphology of lithium anode in Li-S BSBs at different cycle numbers, which corresponds well with EIS results.

Page No. 16 in SI:

Supplementary Figure 14. a-h. EIS of Li-S BSBs at different cycle numbers and their corresponding fitting results. **i.** The equivalent electrical circuit for fitting EIS data after battery cycle and fitting results.

Supplementary Figure 15. The surface morphology of lithium anode in Li-S BSBs at different cycle numbers.

Page No. 16 in MS:

32. Waluś, S., Barchasz, C., Bouchet, R. & Alloin, F. Electrochemical impedance spectroscopy study of lithium–sulfur batteries: Useful technique to reveal the Li/S electrochemical mechanism. *Electrochim. Acta* **359**, 136944 (2020).
33. Shen, X. *et al.* Lithium anode stable in air for low-cost fabrication of a dendrite-free lithium battery. *Nat. Commun.* **10**, 900 (2019).

2. More EIS measurements may be helpful to explain the performance decaying mechanism.

Response:

Thanks for the reviewer's professional review and valuable suggestions. EIS measurements have been used to analyze the potential battery performance decaying mechanism in this revised manuscript.

Page No. 8 in MS:

The lithium anode in Li-S BSBs may affect the battery system's overall performance because there is almost no external pressure on it during battery operation, leading to dendrites growth and electrode surface deterioration. Electrochemical impedance spectroscopy (EIS) analyses were performed on Li-S BSBs at different cycle numbers, as shown in Supplementary Figure 14. The impedance of lithium/electrolyte interphase (R3) increases rapidly as the battery continues to charge/discharge (from 9.57 to 68.97 Ω), indicating that the dendrite growth leads to an overgeneration of solid electrolyte interface (SEI).^{32,33}

Page No. 16 in SI:

Supplementary Figure 14. a-h. EIS of Li-S BSBs at different cycle numbers and their corresponding fitting results. **i.** The equivalent electrical circuit for fitting EIS data after battery cycle and fitting results.

Page No. 16 in MS:

32. Waluś, S., Barchasz, C., Bouchet, R. & Alloin, F. Electrochemical impedance spectroscopy study of lithium-sulfur batteries: Useful technique to reveal the Li/S electrochemical mechanism. *Electrochim. Acta* **359**, 136944 (2020).
33. Shen, X. *et al.* Lithium anode stable in air for low-cost fabrication of a dendrite-free lithium battery. *Nat. Commun.* **10**, 900 (2019).

3. In Fig. 5f, the open circuit voltage of this work was measured at the beginning of the discharge process. I think it is more instructive to exhibit the open circuit voltage at 50% SOC.

Response:

Thanks for the reviewer's professional review and valuable suggestions. We have changed the relevant SOC values in this revised manuscript.

Page No. 23 in MS:

REVIEWER COMMENTS

Reviewer #1 (Remarks to the Author):

I felt that the authors adequately (and more than adequately) addressed my comments.

Reviewer #2 (Remarks to the Author):

I believe that authors revise their article well, and now, this revised version of manuscript is ready to be published in this journal.

Reviewer #3 (Remarks to the Author):

The authors have carefully revised the manuscript. I am happy to see all my questions well answered. I believe the paper is ready for publication.

Reviewer #4 (Remarks to the Author):

This is overall a good work, even though similar idea has been reported in Li-S battery. I suggest the authors to consider our comments below:

1. UV-Vis spectra was used to monitor the concentration of LPS in the top phase during cycling. It was claimed that there was no absorption signal during the charge/discharge which demonstrated that the LPSs cannot shuttle to the top phase. However, the top phase is a DEE-rich phase and as shown in supplementary Figure 11 the LPSs dissolved in DEE have no obvious absorption signal. In this case, no matter whether LPSs exist in the top phase or not, there could be no obvious absorption signal. So, I don't think the UV-Vis spectra results could prove that the LPSs cannot shuttle to the top phase.
2. The potential value for the Li-S BSBs is lower than that of Li-S battery with DOL/DME electrolyte. The authors attribute this potential drop to the dielectric constant difference between electrolytes. I am wondering whether the transfer barrier of Li⁺ between two phases could also contribute to the potential drop. The authors have shown that the Gibbs energy of transfer of Li⁺ from the top phase to the bottom phase is 41.05 kJ/mol which is equal to 0.425 eV. In this case, a potential drop of 0.425 V is introduced in the battery system.
3. In this work, the cathode of Li-S BSBs is cycled between S8 and Li₂S. Since S8 and Li₂S are all solids, technically this is not a flow battery but a new kind of Li-S battery. If the authors hope to compare with other flow batteries, I suggest the authors to provide the performance of Li-S BSBs that barely cycled dissolved polysulfides with no solid formation during the cycling.
4. For redox flow batteries, besides the energy density, people are also quite interested in the power density. Could the authors provide more rate performance of the Li-S BSBs?

Reviewer #4:

This This is overall a good work, even though similar idea has been reported in Li-S battery. I suggest the authors to consider our comments below:

1. UV-Vis spectra was used to monitor the concentration of LPS in the top phase during cycling. It was claimed that there was no absorption signal during the charge/discharge which demonstrated that the LPSs cannot shuttle to the top phase. However, the top phase is a DEE-rich phase and as shown in supplementary Figure 11 the LPSs dissolved in DEE have no obvious absorption signal. In this case, no matter whether LPSs exist in the top phase or not, there could be no obvious absorption signal. So, I don't think the UV-Vis spectra results could prove that the LPSs cannot shuttle to the top phase.

Response:

Thanks for your careful review. In Figure S11, LPSs demonstrated ultra-low solubility in DEE, and deposits of LPSs can be clearly found at the bottom. Therefore, the UV-Vis spectrum did not detect the signal similar to the LPSs dissolved in DMA during BSB operation. In order to more firmly prove that the LPSs cannot shuttle to the top phase, ex-situ Raman spectrum tests were further conducted in revised manuscript. In general, the Raman spectra of LPSs have distinct characteristic peaks in the wavenumber range of 100~500 cm^{-1} (Chem. Soc. Rev., 2019, 48, 5432-5453). We sampled the top phase of Li-S BSBs in different charging/discharging states (Figure S12a). And the corresponding Raman spectrum is summarized in Figure S12b. It can be obviously discovered that the spectra of the top phase electrolyte under various charging/discharging states consisted well with the primary bare electrolyte in the top phase, in which no characteristic peaks of LPS were detected, further indicating that LPSs cannot shuttle from the bottom phase to the top phase during battery operation.

Page No. 7 in MS:

Ex situ Raman tests were conducted to further illustrate the confined ability of LPSs in the biphasic electrolytes. The top phase electrolytes were sampled at various discharge/charge stages (Figure S12a). Their Raman spectra demonstrated a similar trend to the bare top electrolyte, and no characteristic peaks were found between 100-500 cm^{-1} , indicating no LPS shuttle to the top phase,²⁸ and this is consistent well with the in situ UV-Vis spectrometry results.

Page No. 12 in MS:

Raman spectra of electrolytes were conducted with a Horiba Jobin Yvon HR Evolution Raman spectrometer under a Raman laser wavenumber of 633 nm.

Page No. 16 in MS:

28. Zhang, L. *et al.* In situ optical spectroscopy characterization for optimal design of lithium-sulfur batteries. *Chem. Soc. Rev.* **48**, 5432–5453 (2019).

Page No. 14 in SI:

Supplementary Figure 12. The ex situ Raman tests of the top phase electrolyte in Li-S BSBs during discharge/charge.

2. The potential value for the Li-S BSBs is lower than that of Li-S battery with DOL/DME electrolyte. The authors attribute this potential drop to the dielectric constant difference between electrolytes. I am

wondering whether the transfer barrier of Li^+ between two phases could also contribute to the potential drop. The authors have shown that the Gibbs energy of transfer of Li^+ from the top phase to the bottom phase is 41.05 kJ/mol which is equal to 0.425 eV. In this case, a potential drop of 0.425 V is introduced in the battery system.

Response:

Thanks for your careful review and professional comment. The mass transfer resistance of lithium ions at the boundary of the bottom and top phase is largely a key cause of voltage plateau drop. We have added your professional opinion to this revised manuscript.

Page No. 8 in MS:

And as mentioned above, Li^+ also needs to overcome certain energy barriers when crossing the phase boundary during discharge. Therefore, the potential value for the last plateau is relatively lower compared to that in DOL/DME (2 V).

3. In this work, the cathode of Li-S BSBs is cycled between S_8 and Li_2S . Since S_8 and Li_2S are all solids, technically this is not a flow battery but a new kind of Li-S battery. If the authors hope to compare with other flow batteries, I suggest the authors to provide the performance of Li-S BSBs that barely cycled dissolved polysulfides with no solid formation during the cycling.

Response:

Thanks for your professional review. Undoubtedly, the Li-S BSBs that barely cycled dissolved polysulfides can be classified into all-liquid flow batteries. However, Li-S BSBs with insoluble products (S_8 and Li_2S) can be considered as a semi-solid flow battery, a kind of flow battery. Furthermore, the capacity provided by semi-solid process accounts for almost half of the total theoretical capacity and it can be reversible (Nano Letters, 2014, 14, 2210-22180). Its efficient reversibility under dynamic conditions essentially requires further optimization of the battery architecture and design on the current collector to trap solid active particles and converted them effectively, and this is our research emphasis for nonaqueous BSBs in the future. Therefore, it is acceptable to compare the performance of Li-S BSB prototype (including semi-solid process) with conventional flow batteries and current aqueous BSBs. We hope our explanation is acceptable, and thanks again for your instructive comment again.

4. For redox flow batteries, besides the energy density, people are also quite interested in the power density. Could the authors provide more rate performance of the Li-S BSBs?

Response:

Thanks for your valuable comment. The rate performance has been provided in this revised Manuscript (Supplementary Figure 16).

Page No. 8 in MS:

Supplementary Figure 15 shows the rate performance of Li-S BSB, in which the Li-S BSB delivers a specific discharge capacity of 783.3 mAh g⁻¹ even at a high current density of 1 C. The above cycling and rate performance verifies the feasibility of Li-S BSBs.

Page No. 17 in SI:

Supplementary Figure 15. The discharge/charge profiles of Li-S BSB at various current densities.